# Single-molecule analysis of specificity and multivalency in binding of short linear substrate motifs to the APC/C

Nairi Hartooni[1,2], Jongmin Sung[3,4,5], Ankur Jain[3,4,6] & David O. Morgan [1,2✉]

Robust regulatory signals in the cell often depend on interactions between short linear motifs (SLiMs) and globular proteins. Many of these interactions are poorly characterized because the binding proteins cannot be produced in the amounts needed for traditional methods. To address this problem, we developed a single-molecule off-rate (SMOR) assay based on microscopy of fluorescent ligand binding to immobilized protein partners. We used it to characterize substrate binding to the Anaphase-Promoting Complex/Cyclosome (APC/C), a ubiquitin ligase that triggers chromosome segregation. We find that SLiMs in APC/C substrates (the D box and KEN box) display distinct affinities and specificities for the substrate-binding subunits of the APC/C, and we show that multiple SLiMs in a substrate generate a high-affinity multivalent interaction. The remarkably adaptable substrate-binding mechanisms of the APC/C have the potential to govern the order of substrate destruction in mitosis.

[1] Department of Physiology, University of California, San Francisco, CA 94143, USA. [2] Tetrad Graduate Program, University of California, San Francisco, CA 94143, USA. [3] Department of Cellular and Molecular Pharmacology, University of California, San Francisco, CA 94143, USA. [4] Howard Hughes Medical Institute, University of California, San Francisco, CA 94143, USA. [5] Present address: Roche Sequencing Solutions, Santa Clara, CA 95050, USA. [6] Present address: Whitehead Institute for Biomedical Research, Cambridge, MA 02142, USA. ✉email: david.morgan@ucsf.edu

Inside the crowded and noisy confines of the cell, clear and robust regulatory signals require highly specific protein–protein interactions. Many of these interactions depend on the binding of a globular domain in one protein to short linear sequence motifs (SLiMs) in another. SLiMs are short conserved amino acid sequences that are generally found in disordered protein regions, and a remarkably diverse variety of SLiMs is involved in numerous regulatory processes[1]. The affinities and specificities of SLiMs for their targets determine the impact of these motifs in signaling, but we have only a limited understanding of these interactions.

The central importance of SLiM interactions is illustrated by substrate binding to the Anaphase-Promoting Complex/Cyclosome (APC/C)[2]. The APC/C is a conserved 13-subunit ubiquitin ligase that triggers the destruction of key proteins controlling the initiation of chromosome segregation in mitosis[3–6]. Its substrates include the separase inhibitor securin, whose destruction allows separase to separate the duplicated chromosomes. Another key APC/C target is mitotic cyclin, whose destruction is required for late mitotic events. Disordered regions in these substrates contain SLiMs, or degrons, that bind to specific subunits of the APC/C. The APC/C holds the substrate in place while an E2 co-enzyme binds nearby and transfers the ubiquitin to a lysine on the substrate or on ubiquitin. Repeated ubiquitin transfer from multiple E2s leads to the formation of polyubiquitin chains that are recognized by the 26S proteasome, resulting in substrate degradation.

The APC/C is activated in mitosis by one of two related substrate-binding subunits called Cdc20 and Cdh1. These activators contain a globular WD40 domain that binds substrate degrons, flanked by partially disordered regions that mediate binding to the APC/C, resulting in a conformational change that enhances binding of the E2 co-enzyme[7,8]. Activators interact transiently with the APC/C at specific cell cycle stages. Cdc20 activates the APC/C during metaphase and anaphase of mitosis and binds a narrow range of substrates governing the initiation of chromosome segregation[9]. In late anaphase, Cdc20 is replaced by Cdh1, which activates the APC/C in late mitosis and G1[9]. Cdh1 has broader specificity and targets many additional proteins for destruction.

Three major degrons have been identified in APC/C substrates: the destruction box (D box), KEN box, and ABBA motif[2,10–16] (Supplementary Fig. 1a). As with most SLiMs, these degrons are found in disordered regions, and substrates often contain multiple degrons. The most important degron is the D box, which has a composite binding site involving both the WD40 domain of the activator and the Apc10/Doc1 subunit of the APC/C. The conserved residues of the D box are RxxLxxxxN. The N-terminal RxxL segment interacts with an acidic patch and aliphatic pocket on the WD40 domain of the activator[10]. The C-terminal residues of the D box interact with the Apc10 subunit[16–18]. As a result, the D box helps anchor the activator to the APC/C[19,20]. The second major APC/C degron, often found near a D box, is the KEN box, which usually contains a well-conserved KEN sequence that interacts with a specific binding pocket on the activator WD40 domain[2,10]. Lastly, the less common ABBA motif has a complex consensus sequence that interacts with a specific groove on the activator WD40 domain[2,11]. In yeast, variations in this motif result in specificity for one or the other activator[2,11,15].

APC/C substrates are targeted for destruction in a specific order during mitosis. Substrates that control anaphase onset, such as securin, are generally degraded earlier, in metaphase, than substrates involved in late mitotic events. This order is likely to be achieved in part by the selectivity of the activators for different substrates. Destruction of securin and a small number of other early substrates depends on Cdc20, whereas numerous later substrates, degraded in late mitosis and G1, are targeted specifically by Cdh1[19,21]. There is some evidence for activator-specific D-box sequences, as well as evidence that the KEN box has a preference for Cdh1[12]. However, activator specificity alone cannot explain all substrate ordering. The same activator is known to target different substrates at different times, perhaps due to variations in degron affinity, combinations of multiple degrons, or other mechanisms[2,15,22,23].

Substrate affinity for the APC/C is a critical determinant of the extent of ubiquitylation. As a ubiquitin ligase of the RING family, the APC/C binds substrates at one site while the E2-ubiquitin conjugate binds at a nearby site, enabling lysines in the disordered substrate to attack the E2 to catalyze transfer[3,4,6]. Ubiquitylation is processive: multiple E2-ubiquitin conjugates can bind, transfer ubiquitin, and dissociate during a single substrate-binding event[24–27]. Thus, the number of ubiquitins added is directly dependent on substrate dwell time, or dissociation rate. It is likely that proteasome recognition depends on the number and length of polyubiquitin chains, so different substrate dwell times are likely to influence the timing of their degradation[22,28].

Despite decades of research on the APC/C, the affinity of substrate binding remains poorly understood. Conventional approaches to affinity analysis have been hampered by our inability to express and purify large amounts of the multi-subunit APC/C or its activators. To solve this problem, we developed a single-molecule binding assay that provides robust measurements of the rate of dissociation of substrates bound to activators and the activated APC/C. These methods provide important new insights into degron affinity, activator specificity, and multivalency. Our methods can also be applied to binding interactions with other proteins and protein complexes that are not readily studied by conventional ensemble methods.

## Results

**Analysis of degron affinity by ensemble biochemistry**. To determine the affinities of APC/C degrons for their binding sites, we first used conventional equilibrium binding assays of degron peptide binding to the activator. We used the baculovirus system to produce the WD40 domain of Cdh1 from the budding yeast *Saccharomyces cerevisiae* and measured fluorescence anisotropy to assess the binding of fluorescently-tagged degron peptides at increasing activator concentrations. The WD40 domain of yeast Cdc20 could not be expressed and was not studied.

Our studies centered on the well-known D box of the yeast protein Hsl1 (Fig. 1a), a late mitotic substrate that is targeted primarily by APC/C[Cdh1] in vivo[29]. The Hsl1 D box is known to have an ideal consensus sequence that binds tightly to APC/C[Cdh1], resulting in highly processive modification[19,30]. We found that the Hsl1 D box binds the Cdh1 WD40 domain with a dissociation constant ($K_D$) of 3.5 μM (Fig. 1a, b). Binding was abolished by the mutation of three key residues in the D box.

We also analyzed the D boxes of yeast securin/Pds1(ySecurin) and the S-phase cyclin Clb5. These proteins are targeted by APC/C[Cdc20] prior to anaphase in vivo but are also thought to be modified by APC/C[Cdh1] in G1[15]. Neither D box displayed significant binding to Cdh1 in our assay, suggesting that these D boxes have a very low affinity for this activator (Fig. 1a, b).

We also tested the KEN boxes of Hsl1 and ySecurin. In both cases, binding saturation was not achieved at the highest concentration of Cdh1 (30.6 μM), reducing the accuracy of $K_D$ values. The results yielded $K_D$ estimates of 14 and 26 μM for the Hsl1 and ySecurin KEN degrons, respectively (Fig. 1a, b). We also tested a Cdh1-specific ABBA motif from the pseudosubstrate yeast protein Acm1. This motif bound with very low affinity to Cdh1 (Supplementary Fig. 1b).

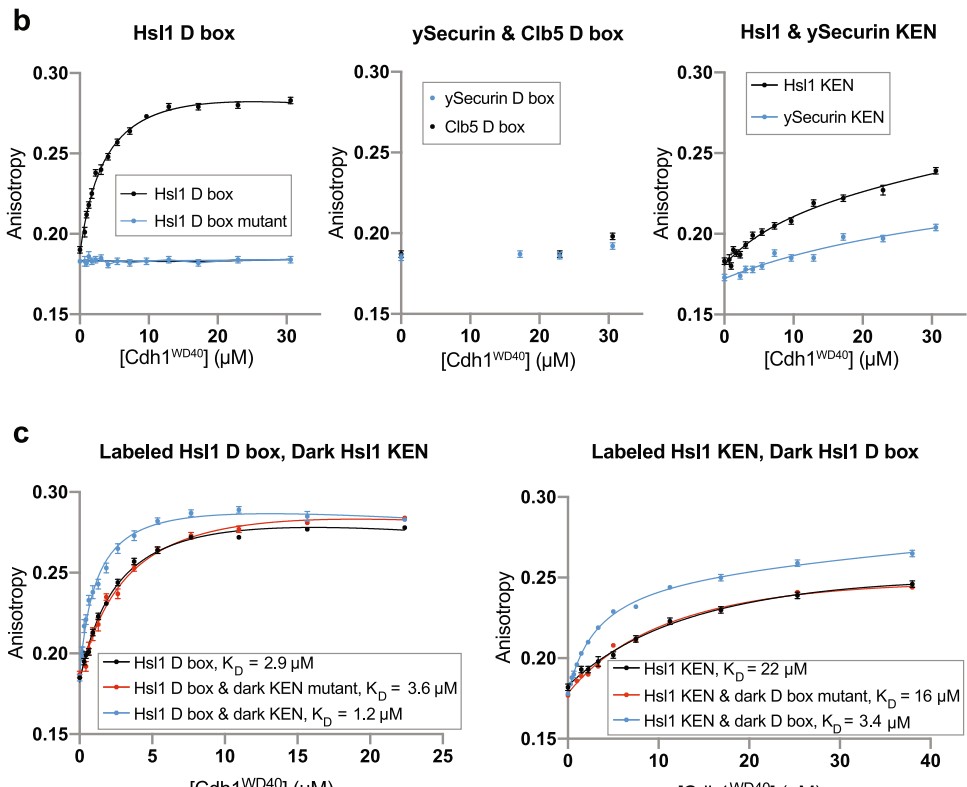

| Degron | Peptide Sequence | $K_D$ (µM; Mean ± SE) |
|---|---|---|
| Hsl1 D box | EQKPK**R**AA**L**SDIT**N**SFNKMN-K(Cy5) | 3.5 ± 0.32 (n = 3) |
| Hsl1 D box mutant | EQKPK**A**AA**A**SDIT**A**SFNKMN-K(Cy5) | >> 30.6 |
| ySecurin D box | AQQQG**R**LP**L**AAKD**N**NRSKSFI-K(Cy5) | >> 30.6 |
| Clb5 D box | QDSKP**R**RA**L**TDVPV**N**NNPLSQ-K(Cy5) | >> 30.6 |
| Hsl1 KEN | GVSTN**KEN**EGPEYPTKIE-K(Cy5) | 14 ± 4.6 (n = 3) |
| ySecurin KEN | PANED**KEN**NIVYTG-K(Cy5) | 26 ± 14 (n = 2) |
| Hsl1 D box + Dark Hsl1 KEN | | 1.1 ± 0.8 (n = 2) |
| Hsl1 D box + Dark Hsl1 KEN mutant | | 2.9 ± 0.7 (n = 2) |
| Hsl1 KEN + Dark Hsl1 D box | | 3.4 ± 0.3 (n = 2) |
| Hsl1 KEN + Dark Hsl1 D box mutant | | 11 ± 5 (n = 2) |

**Fig. 1 Analysis of degron affinity by fluorescence anisotropy. a** List of Cy5-labeled degron peptides tested in anisotropy experiments with Cdh1^WD40. Column at right lists mean $K_D$ values from multiple experimental replicates, including the representative experiments shown in **b** and **c**. **b** Fluorescence anisotropy experiments with the peptides listed in **a**. 10 nM peptide was incubated with up to 30.6 µM Cdh1^WD40. Data points represent mean ± SE (n = 10 reads per reaction). In this experiment, $K_D$ values were as follows: Hsl1 D box, 4 µM; Hsl1 KEN, 12 µM; ySecurin KEN, 40 µM. Mean $K_D$ values from 2 or 3 independent experiments are shown in **a**. **c** Binding was measured with labeled degron peptide (10 nM) in the absence (black) or presence of an unlabeled version of the other degron (100 µM), either wild-type (blue) or mutated at key residues (red). Note that 100 µM KEN peptide is not completely saturating. Data points represent mean ± SE (n = 10 reads per reaction). Insets indicate $K_D$ values in this experiment. Mean $K_D$ values from two independent replicates are shown in **a**. Source data are provided as a Source Data file.

Substrates of the APC/C often contain both a D box and a KEN box, generally at a distance that should allow simultaneous binding. We tested the possibility that the binding of one degron affects affinity for the other. The addition of unlabeled Hsl1 KEN peptide improved affinity for the Hsl1 D box about 3-fold (Fig. 1a, c). As expected for an allosteric mechanism, the reverse was also true: saturating D box peptide improved affinity for the KEN box about 4-fold (Fig. 1a, c).

**APC/C single-molecule assay development**. Conventional assays like that used in Fig. 1 are limited by the need for very large amounts of the purified binding protein, which is possible for the Cdh1 WD40 domain but not possible for Cdc20 or for the APC/C or APC/C-activator complexes. To thoroughly probe the interaction of substrates with the APC/C, we developed a Single

Molecule Off Rate (SMOR) assay in which dynamic substrate-APC/C interactions can be visualized and quantified by fluorescence microscopy. Our goal was to create an adaptable and simple-to-use platform that could be deployed to probe protein–protein interactions for any protein or protein complex that cannot be purified in large quantities.

We applied a previously developed antibody-based method to tether single protein molecules on a functionalized glass surface[31], and modified it to capture transient interactions with fluorescent ligands using total internal reflection fluorescence (TIRF) microscopy.

NeutrAvidin and biotinylated antibodies were used to tether molecules to glass coverslips in small chambers with ports for influx and outflow of ligand solutions[31]. We populated the surface with budding yeast activator (Cdh1^WD40), activated APC/C (APC/C^Cdh1

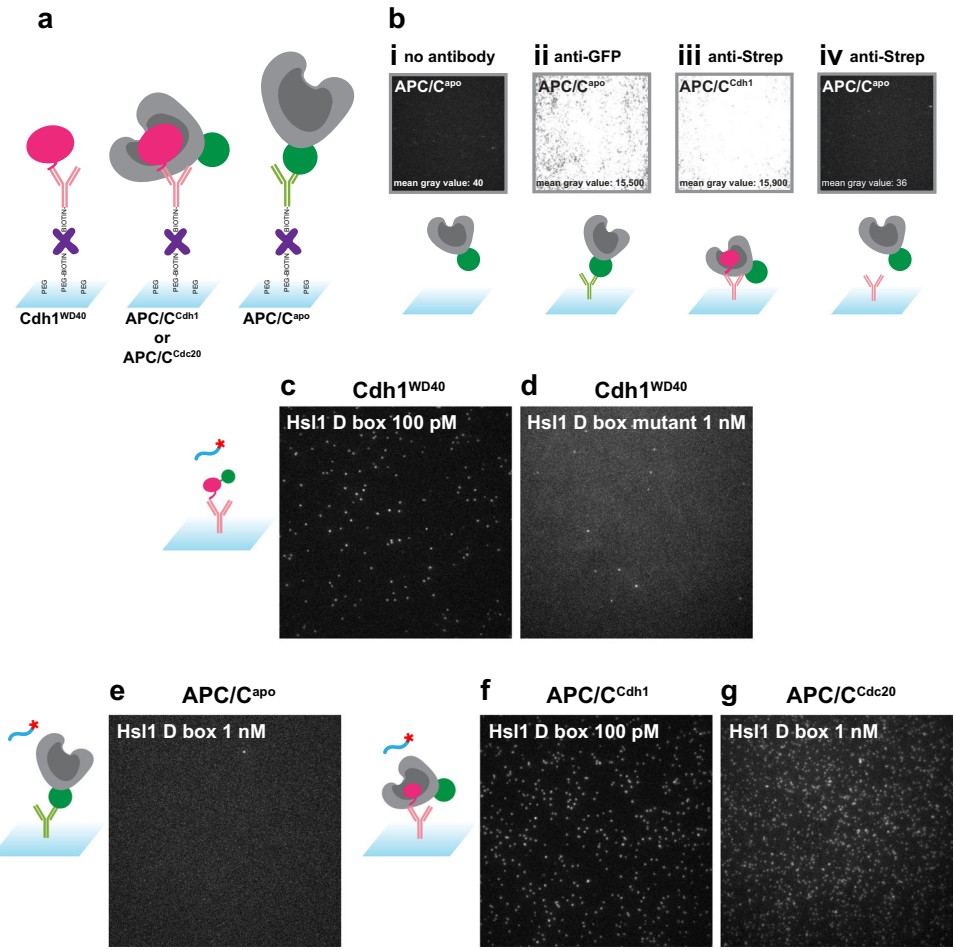

**Fig. 2 SMOR assay setup. a** Immobilization of binding proteins on the cover glass surface. For activator or for activated APC/C, we used biotinylated anti-Strep-Tag II antibody (pink), which binds the Strep-Tag II on the activator N-terminus. For APC/C$^{apo}$, we used biotinylated anti-GFP antibody (green), which binds the GFP tag on the Apc1 subunit. Antibody is linked to the biotinylated PEG on the surface using NeutrAvidin (purple). **b** Fluorescent signal from APC/C-GFP in the absence and presence of antibodies and activator as indicated. **c, d** Single-molecule interactions of Cy5-labeled Hsl1 D box peptide (**c**) or mutant peptide (**d**) with immobilized GFP-tagged Cdh1$^{WD40}$. **e** Lack of interactions between Hsl1 D box peptide and immobilized APC/C$^{apo}$. **f, g** Single-molecule interactions of Cy5-labeled Hsl1 D box peptide with immobilized APC/C$^{Cdh1}$ (**f**) or APC/C$^{Cdc20}$ (**g**). Images in **c–g** are maximum intensity projections of the first 10 frames of a movie at continuous exposure and 100 ms frame rate (Supplementary Movies 1–5).

or APC/C$^{Cdc20}$), or APC/C lacking activator (APC/C$^{apo}$). Activator proteins were produced with the baculovirus system, APC/C was purified from yeast cells, and APC/C-activator complexes were prepared by mixing purified APC/C and activator prior to immobilization on the glass (Supplementary Fig. 2a). Very small amounts of protein were required. Typically, excellent glass coverage could be achieved with less than a nanogram of protein.

To confirm the successful capture of the target molecule on the glass surface, C-terminal GFP tags were fused to the Cdh1$^{WD40}$ protein and the Apc1 subunit of the APC/C. The C-terminal Apc1 tag did not affect APC/C ubiquitylation activity in vitro (Supplementary Fig. 2b). Activators were N-terminally tagged with a Strep Tag II, which had no effect on ubiquitylation activity in vitro (Supplementary Fig. 2c). Proteins were immobilized on glass using either biotinylated anti-Strep Tag II antibody to bind activator or anti-GFP antibody to bind the GFP-tagged APC/C$^{apo}$ (Fig. 2a).

GFP fluorescence was not observed when there was no antibody immobilized on the surface, indicating that our glass functionalization scheme minimized background APC/C binding (Fig. 2bi). In contrast, anti-GFP antibody specifically immobilized abundant APC/C$^{apo}$ (Fig. 2bii). Similarly, anti-Strep Tag II antibody specifically immobilized activated APC/C, and no cross-reactivity with APC/C$^{apo}$ was observed (Fig. 2biii, iv). Note that

immobilization of activated APC/C with a tag on the activator subunit ensures that we are measuring interactions only with intact APC/C that retains activator binding activity.

We carried out initial binding studies with the same Cy5-labeled Hsl1 D box peptide that we used for our binding analysis in Fig. 1. Capturing the signal from the Cy5 dye by TIRF microscopy, we observed binding to Cdh1$^{WD40}$ at 100 pM peptide (Fig. 2c and Supplementary Movie 1), and very little binding with just antibody on the surface (Supplementary Fig. 3). The mutant D-box peptide displayed negligible binding (Fig. 2d and Supplementary Movie 2). Furthermore, the Hsl1 D box peptide did not interact with APC/C$^{Apo}$ or the anti-GFP antibody used to tether it to the surface (Fig. 2e and Supplementary Fig. 3, Supplementary Movie 3). Although the Apc10 subunit of the APC/C is believed to interact with the C-terminal residues of the D box, the affinity of this interaction is known to be extremely low. Finally, we demonstrated that the Hsl1 D box peptide interacts with APC/C activated with either Cdh1 or Cdc20 (Fig. 2f, g and Supplementary Movies 4, 5).

**Computational analysis of ligand dwell time.** To quantify the affinity of substrate-APC/C interactions, we next developed data

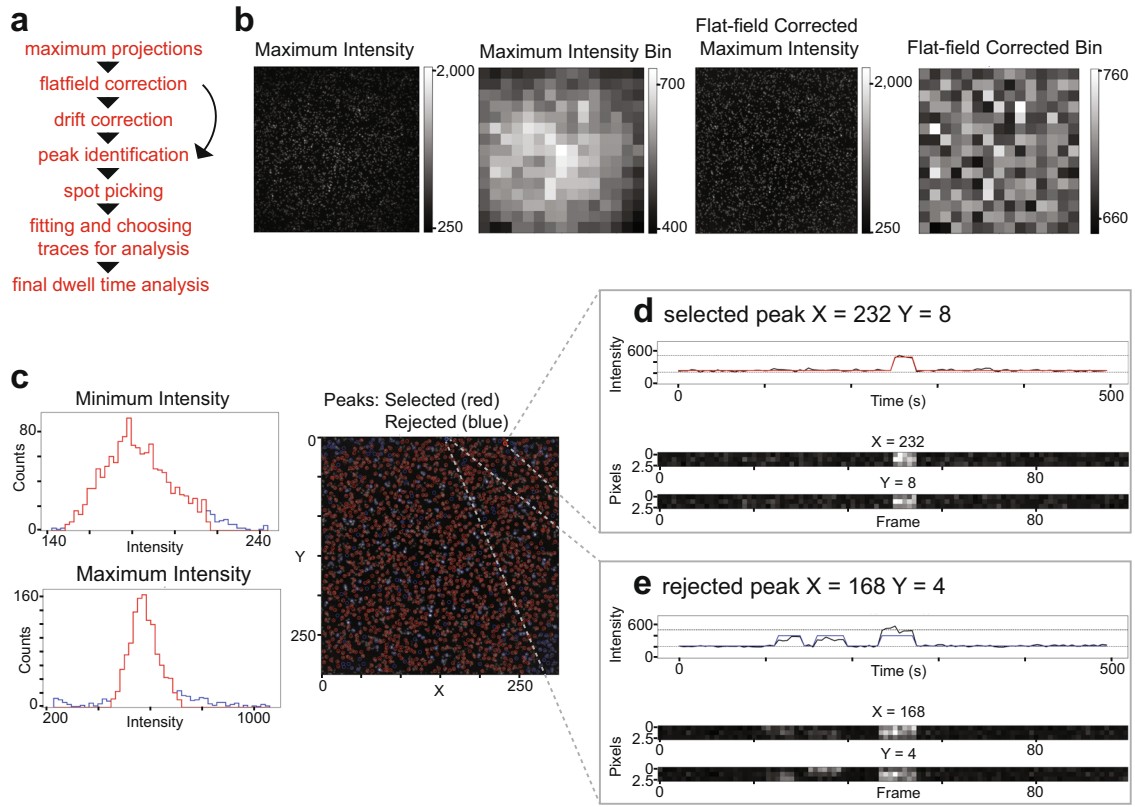

**Fig. 3 SMOR analysis. a** Analysis pipeline for SMOR analysis. **b** For flat-field correction, the maximum intensity projection of a 300 × 300 pixel movie is binned into a 20 × 20 pixel grid, flat-field corrected, and then binned again to check correction. **c** Histograms of minimum and maximum intensity values along traces for each peak. Red indicates selected peaks (≤3 SD from median) and blue indicates rejected peaks. At right is a maximum intensity projection with selected (red circle) and rejected (blue circle) peaks (*x, y* coordinates). **d** Representative single-molecule trace of a selected peak at coordinates (232, 8). **e** Representative single-molecule trace of a rejected peak at coordinates (168, 4).

analysis methods to determine the length of time that a fluorescent ligand remains bound to its binding partner on the glass surface. The reciprocal of the mean dwell time is the dissociation rate constant, $k_{\text{off}}$, which provides important clues about the extent of multiubiquitylation by the APC/C, and thus the substrate degradation rate in the cell[32].

Signal intensity in single-molecule studies depends on the nature of the dye, the parameters of the microscope, and whether the light being captured is from a monomeric molecule or a much brighter multimer. Our analysis pipeline is designed to account for all these factors for both short and long binding events and is robust to experimental and technical perturbations. In short, the pipeline corrects movies for different intensities across the field of view, corrects for drift if needed[33], and then analyzes information on signal intensity to identify single-molecule binding events and calculate dwell time (Fig. 3a). Furthermore, oxygen scavenging agents were used in all experiments, ensuring that photobleaching occurred over much longer time scales than observed dwell times.

Figure 3 illustrates our analysis methods using the binding of Cy5-labeled Hsl1 D box peptide to Cdh1-activated APC/C. First, the 512 × 512 pixel movie was cropped to the central 300 × 300 pixel grid. Analysis of signal intensity across the grid was then used to generate image projections of maximum and minimum intensities at all pixels during the length of the video (Fig. 3b and Supplementary Fig. 4a). The minimum intensity projection reveals a low level of long-lived nonspecifically bound fluorescent substrate on the glass. The maximum intensity projection shows potential transient binding events. In this example, there was a 10-fold difference between the range of intensities found in the maximum and minimum intensity projections (Supplementary

Fig. 4a). Greater differences in intensity between the two indicate higher signal-to-noise ratio and thus more robust detection of binding events.

Binding signals in the maximum intensity projection tend to be brighter at the center of the TIRF evanescent wave, which can be a problem as we use the intensity level of the fluorescence signal to identify single molecules and discard multimers. To apply flat-field correction, intensities from the maximum intensity projection were used to create a mask of areas with any signal (Supplementary Fig. 4b). After the application of the mask, the average intensities in 20 × 20 pixel grids were used to create an intensity bin image that shows the center of the TIRF evanescent wave (Fig. 3b). The edges of the bins were smoothened with a Gaussian filter to obtain an intensity bin filter, which was used to normalize intensities across the grid for flat-field correction (Supplementary Fig. 4b). To confirm flat-field correction, the same process was repeated on the corrected image, revealing more evenly distributed illumination (Supplementary Note 1). The corrected dataset was then applied to the next step, which for longer acquisition intervals includes drift correction (Supplementary Fig. 4c). Peak intensities or "peaks" were identified as 3 × 3 pixel squares (1 pixel = 16 × 16 µm) centered on (*x, y*) coordinates on this corrected maximum intensity projection (Supplementary Fig. 4d).

Next, we identified peaks that were most likely to represent genuine binding events. The first step was to discard peaks that were too bright, indicating a multimer. We created histograms of minimum and maximum intensities at each peak on the 300 × 300 pixel grid (Fig. 3c). Multimers were excluded in most cases by discarding peaks that were three standard deviations

above the median maximum intensity. The same was done for minimum intensity peaks to eliminate background noise from dimmer signals.

For each selected peak, the signal intensity trace over time was fit to a Hidden Markov Model (HMM)[34] to determine a bound/ unbound state trajectory. First, the intensities throughout the movie along single-molecule traces at all peaks were plotted in a histogram and fit using a double Gaussian to determine the mean maximum and minimum intensity values for the overall signal unique to each movie (Supplementary Fig. 4e). These initial parameters were used in the HMM to define bound and unbound states (Supplementary Fig. 4f, g). Deviation of the intensity data from the HMM fit for each trace was calculated using root mean squared deviation (RMSD). A histogram of RMSD values for HMM fitting of all traces was created, and a trace was rejected if the RMSD value was more than two standard deviations away from the median (Supplementary Fig. 4h). If a trace fell within the intensity parameters and had a low RMSD value, it was included in the analysis and colored red (Fig. 3d). If a trace had a high RMSD value, it was not included and colored blue. In the example in Fig. 3e, fluorescence at a nearby binding event created deviations in intensity during the movie and resulted in a higher RMSD value.

For each peak selected as a genuine binding event in the HMM, we used maximum likelihood estimation with an exponential distribution to statistically infer the dwell time. For each movie, a histogram showing the distribution of dwell times from multiple traces was calculated from the estimated inverse cumulative density function (Fig. 4).

The minimum frame rate of our camera with full use of all active pixels was 32 ms. Ideally, the calculated mean dwell time should be greater than three times the frame rate, and thus we were unable to reliably measure dwell times less than 100 ms. There were multiple instances in which we observed single-frame interactions. Although a dwell time could not be calculated in these cases, they are likely to represent real binding and are noted in our analysis as 'single-frame' events.

We performed multiple independent experiments for each ligand-protein combination. A single representative replicate for each condition is described in the following sections. Additional replicates are listed in Supplementary Table 1.

**Cooperation between activator and Apc10 in D box binding**. We first quantified the binding of the Hsl1 D box peptide to Cdh1 and to APC/C-activator complexes. The mean dwell time for the peptide with the Cdh1 WD40 domain was $0.360 \pm 0.005$ s (Fig. 4a, Table 1; note that the error in these analyses is an estimated standard error of the mean for an exponential distribution). The dissociation rate constant $k_{off}$ for this interaction is therefore $2.8$ s$^{-1}$. Based on the $K_D$ of $3.5 \times 10^{-6}$ M that we determined earlier (Fig. 1c), we infer an association rate constant $k_{on}$ of $8 \times 10^5$ M$^{-1}$ s$^{-1}$, which is at the low end of the normal range of diffusion-limited binding events[35].

The dwell time of Hsl1 peptide binding to Cdh1-activated APC/C was $35.3 \pm 1$ s (Fig. 4b and Table 1). This ~100-fold increase in affinity relative to Cdh1 alone seemed likely to be due to the presence of the Apc10 subunit of the APC/C, which interacts with the C-terminal end of the D box (Supplementary Fig. 1a). We tested this possibility with purified APC/C contain- ing a mutant Apc10 subunit, Apc10-4A, that contains four point mutations that eliminate D-box binding[30]. As predicted, these mutations resulted in a ~100-fold decrease in mean dwell time to $0.375 \pm 0.01$ s, which is roughly equal to the dwell time with Cdh1 alone (Fig. 4c and Table 1).

The affinity of the D box for Apc10 is known to be extremely low, as confirmed by the lack of detectable D-box binding to

APC/C$^{apo}$ (Fig. 2e). Moreover, we did not observe detectable binding of the Hsl1 D box to $144 \mu$M purified Apc10 in anisotropy experiments (Supplementary Fig. 5a). We conclude that a weak interaction with Apc10 cooperates with Cdh1 to provide high-affinity D-box binding. If we assume that $k_{on}$ for D-box binding to APC/C$^{Cdh1}$ is the same as that for binding to Cdh1 alone, then we would estimate a $K_D$ of 36 nM for the binding of the Hsl1 D box to APC/C$^{Cdh1}$.

To confirm that the APC/C$^{apo}$ in these experiments was functional, we also tested a Cy5-labeled peptide of the C-terminal IR motif of Cdh1, which binds to the Cdc27 subunit[4]. We recovered specific protein–protein interactions with a dwell time of $0.616 \pm 0.02$ s (Supplementary Fig. 5b, c).

The Hsl1 D box bound Cdc20-activated APC/C with a mean dwell time of $0.412 \pm 0.007$ s (Fig. 4d), 85-fold lower affinity than that for APC/C$^{Cdh1}$. Thus, the Hsl1 D box has a clear preference for Cdh1, which is consistent with the evidence in vivo that Hsl1 is primarily a Cdh1 target late in mitosis. The Apc10-4A mutation reduced dwell time to $0.152 \pm 0.002$ s (Fig. 4e). This 3-fold drop in dwell time is far less dramatic than the 100-fold decrease seen with APC/C$^{Cdh1}$, perhaps suggesting that the activator influences the ability of the D box to interact with Apc10; that is, the Hsl1 D box peptide does not engage with the Apc10 subunit in the same way when bound to Cdc20.

In previous studies with purified yeast components, Cdh1 dissociation from the APC/C was not detected in 30 min, and activator affinity was enhanced by D-box binding[19,20]. It is therefore unlikely that activator dissociation is occurring during our measurements of substrate dissociation from APC/C$^{Cdh1}$ or APC/C$^{Cdc20}$.

**Activator specificity of degrons**. We next used the SMOR assay to analyze the other Cy5-labeled degron peptides used in our anisotropy studies (Fig. 1a). In contrast to the D box of Hsl1, the ySecurin D box displayed specificity for APC/C$^{Cdc20}$. Mean dwell time with APC/C$^{Cdc20}$ was $1.87 \pm 0.04$ s, compared with a mean dwell time with APC/C$^{Cdh1}$ of $0.132 \pm 0.002$ s (Fig. 5a). Despite being one of the earliest APC/C$^{Cdc20}$ substrates in vivo, we found that the D box of Clb5 had a similar affinity for APC/C$^{Cdc20}$ ($0.207 \pm 0.008$ s) and APC/C$^{Cdh1}$ ($0.159 \pm 0.004$ s) (Fig. 5b). It is likely that the early degradation of Clb5 relative to securin depends not on D box selectivity but on the presence of a Cdc20- specific ABBA motif in Clb5[15], as well as other mechanisms that promote Clb5 binding to the APC/C (see Discussion).

The KEN peptide from Hsl1 bound Cdh1 and APC/C$^{Cdh1}$ with similar affinity (Fig. 5c), consistent with the idea that this degron binds to the activator and not to other APC/C subunits. The KEN peptide from ySecurin bound only transiently to Cdh1 (single- frame events), suggesting a low affinity. There was no detectable binding of either KEN peptide to APC/C$^{Cdc20}$ (Table 1).

We also analyzed interactions between human activators and degrons. We were able to prepare bulk quantities of the WD40 domains of Cdc20 and Cdh1 for fluorescence anisotropy studies. We observed good binding ($K_D \sim 2 \mu$M) to both activators by the Hsl1 D box and significant but low-affinity binding to Cdh1, but not Cdc20, by the KEN box from human securin/Pttg1 (hSecurin) and D box from Aurora B kinase (AurKB) (Supplementary Fig. 6). We also used single-molecule studies to analyze the binding of various degrons to human activators (Table 2). The hSecurin KEN peptide did not bind human Cdc20$^{WD40}$ but bound human Cdh1$^{WD40}$ with a dwell time of $0.174 \pm 0.003$ s (Fig. 5d, Supplementary Fig. 7, and Table 2). Similarly, the AurKB D box bound human Cdh1$^{WD40}$ with a dwell time of $0.202 \pm 0.003$ s (Fig. 5e and Supplementary Fig. 7) but displayed a lower affinity for Cdc20 (single-frame interactions only).

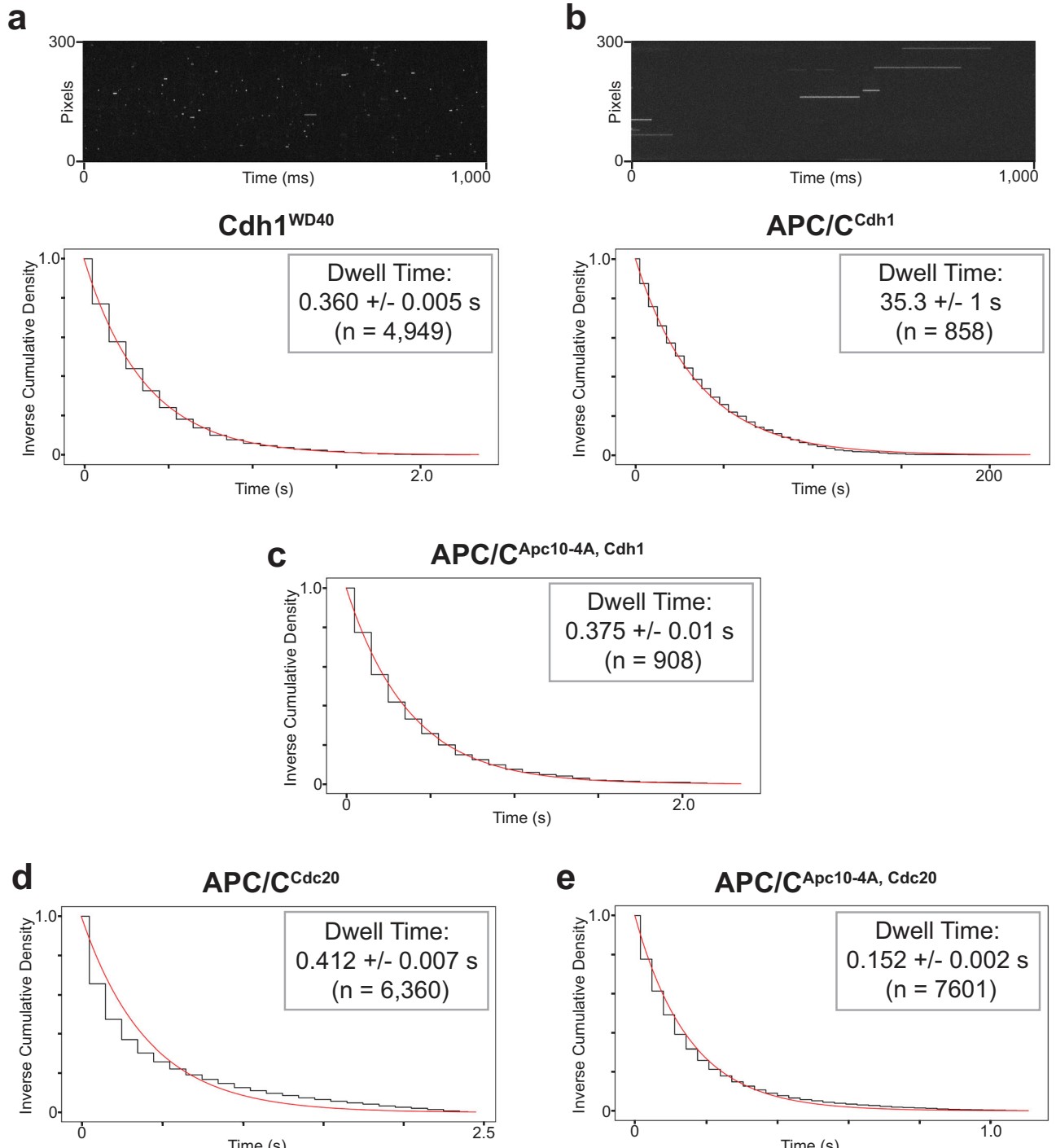

**Fig. 4 Hsl1 D box binding analysis.** Dwell time distributions from SMOR analysis of representative movies with Cy5-labeled Hsl1 D box peptide and GFP-tagged binding protein on the glass surface: **a** Cdh1$^{WD40}$ (100 pM peptide); **b** APC/C$^{Cdh1}$ (100 pM peptide); **c** APC/C$^{Cdh1}$ with Apc10-4A mutations (1 nM peptide); **d** APC/C$^{Cdc20}$ (1 nM peptide); **e** APC/C$^{Cdc20}$ with Apc10-4A mutations (1 nM peptide). Panels **a** and **b** include kymographs of binding events over time. Insets indicate mean dwell time ± SE (*n* indicates number of selected peaks). Results are representative of two independent experiments (Supplementary Table 1).

Attempts to study a human securin D box peptide were not successful due to peptide aggregation and high background.

These experiments revealed that yeast degrons (Hsl1 D box and KEN, ySecurin KEN) bind human Cdh1 with ~10-fold longer dwell times than they bind yeast Cdh1 (Tables 1 and 2). However, the $K_D$ values for Hsl1 D box-binding are similar for the two species (Fig. 1a and Supplementary Fig. 6b), and the inferred $k_{on}$ for D-box binding is ~8-fold lower for human Cdh1 ($10^5$ M$^{-1}$ s$^{-1}$

for human, $8 \times 10^5$ M$^{-1}$ s$^{-1}$ for yeast). Thus, $k_{on}$ might not be simply diffusion-limited but is influenced by other factors such as electrostatics or conformational changes.

**Substrate with multiple degrons binds with very high affinity.** APC/C substrates often contain multiple degrons. To fully understand substrate interactions with the APC/C, we analyzed

**Table 1 Dwell times for substrate interactions with yeast APC/C.**

| | Cdh1[WD40] | APC/C[Cdh1] | APC/C[Cdc20] | APC/C[Apc10-4A, Cdh1] | APC/C[Apc10-4A, Cdc20] |
|---|---|---|---|---|---|
| Hsl1 D box | 0.360 s | 35.3 s | 0.412 s | 0.375 s | 0.152 s |
| Hsl1 D box mutant | No binding | No binding | No binding | No binding | No binding |
| ySecurin D box | SF | 0.132 s | 1.87 s | SF | SF |
| Clb5 D box | SF | 0.159 s | 0.207 s | | |
| Hsl1 KEN | 0.166 s | 0.185 s | No binding | | |
| ySecurin KEN | SF | SF | No binding | | |
| Acm1 ABBA | SF | No binding | No binding | | |
| Hsl1[Halo] KEN & D | 56.4 s | 321 s | 0.518 s | 75.9 s | 0.449 s |
| Hsl1[Halo] KEN & ΔD | 0.421 s | 0.374 s | No binding | | |
| Hsl1[Halo] ΔKEN & D | 3.10 s | 41.6 s | SF | 3.31 s | SF |
| Hsl1[Halo] ΔKEN & ΔD | No binding | No binding | No binding | No binding | No binding |
| AurKB D box | SF | No binding | No binding | | |
| hSecurin KEN | No binding | No binding | No binding | | |

Dwell times are representative of two independent replicates, listed in Supplementary Table 1.
SF single frame.

its interaction with a substrate carrying both D box and KEN degrons. We used a well-studied fragment of Hsl1 (amino acids 667–872)[29], tagged with a C-terminal HaloTag to which chemical dye JF549 covalently binds[36] (Fig. 6a). APC/C ubiquitylation assays and an APC/C ensemble binding assay confirmed that the HaloTag does not affect ubiquitylation or binding (Supplementary Fig. 2d, e). This substrate, including mutants lacking one or both degrons, was tested under the same conditions as those in our peptide binding experiments and found to have specific single-molecule binding (Supplementary Fig. 8). The dwell times are summarized in Table 1.

The combination of both a KEN and D box in a single substrate resulted in extremely high-affinity binding. The dwell time with Cdh1[WD40] increased 100-fold from ~0.4 and ~0.2 s for D and KEN box peptides, respectively, to 56.4 ± 2 s for the Hsl1 fragment (Fig. 6b). The interaction was increased another 6-fold with APC/C[Cdh1] (dwell time 321 ± 14 s; Fig. 6c). This boost in affinity was not seen in the Apc10-4A mutant (Fig. 6d), as in our earlier studies of the D box alone (Fig. 4c). Again, applying the $k_{on}$ estimated from anisotropy experiments, we infer that the $K_D$ of Hsl1 for APC/C[Cdh1] is ~4 nM.

Activator specificity was retained by the Hsl1 fragment, as the dwell time of 0.518 ± 0.03 s with APC/C[Cdc20] is 600-fold lower than that with APC/C[Cdh1] (Fig. 6e). This specificity can also be seen in the processivity of ubiquitylation of this substrate (Supplementary Fig. 2f). Interestingly, as seen in our studies of the Hsl1 D box peptide, the Apc10-4A mutations only slightly reduced Hsl1 dwell time with APC/C[Cdc20] (Table 1), suggesting as before that the Hsl1 D box does not engage effectively with Apc10 when bound to Cdc20.

An Hsl1 fragment carrying mutations in the D box was also tested. We would expect this substrate to interact primarily through the KEN box. This substrate displayed a recoverable dwell time with Cdh1[WD40] (0.421 ± 0.008 s) and APC/C[Cdh1] (0.374 ± 0.02 s) but not with APC/C[Cdc20] (Fig. 6f, g and Table 1). These dwell times are very similar to those observed with the KEN peptide, further suggesting that the KEN motif binds poorly to Cdc20.

We also tested an Hsl1 fragment with mutations in the KEN box. This substrate was expected to bind primarily through the D box. Interestingly, we found a dwell time of 3.10 ± 0.07 s with Cdh1[WD40], a 10-fold increase from that with the D box peptide alone (Fig. 6h and Table 1). The interaction of this mutant substrate with APC/C[Cdh1] was similar to that with the D box peptide (Fig. 6i). To test if the 10-fold increase with Cdh1 alone was due to Hsl1[Halo] interacting with a portion of the WD40 domain that is occluded by the APC/C, we also tested this mutant

with Cdh1-activated APC/C[Apc10-4A]. This interaction occurred with a dwell time of 3.31 ± 0.06 s, which is similar to that with Cdh1[WD40] (Fig. 6j). These results suggest that an Hsl1 sequence outside the tested D box peptide interacts with Cdh1 in a way that is blocked by the interaction with Apc10.

## Discussion

Quantitative analysis of macromolecular interactions is often hindered by the low protein yields that result from heterologous overexpression and purification. We addressed this problem by developing a straightforward, adaptable, and robust single-molecule binding assay that requires minimal amounts of protein (nanograms). We used the SMOR method to carry out quantitative analyses of APC/C interactions with its substrates, providing new insights into the specificity of substrate binding for different activators, and the role of multivalent degron interactions in high-affinity binding.

The activators of the APC/C are thought to possess distinct substrate specificities[15,21]: Cdc20 triggers anaphase by promoting degradation of a small number of substrates (including securin and Clb5), while in late mitosis and G1 Cdh1 promotes degradation of an expanded range of substrates (including Hsl1). Ubiquitylation of securin and Clb5 by APC/C[Cdc20] is more processive than that with APC/C[Cdh1], while Hsl1 is more processively modified by APC/C[Cdh1] [19]. We now provide direct quantitative evidence to demonstrate activator specificity in degron binding. We find that the yeast securin D box displays 15-fold higher affinity for APC/C[Cdc20], while the Hsl1 D box has 85-fold preference for APC/C[Cdh1]. These preferences presumably depend on residues in the degron other than the conserved RxxL consensus, which interact with specific features of the binding site on the activator.

Differences in the D-box binding site of the activators are further supported by the Cdc20 specificity of the chemical inhibitor Apcin[37]. This specificity raises the exciting possibility of activator-specific targeted protein degradation as a therapeutic application.

Surprisingly, the D box of Clb5 displays a similar (moderate) affinity for both activators despite its preference for Cdc20 in vivo and in ubiquitylation assays. It seems likely that Cdc20 specificity in the case of Clb5 is provided by its ABBA motif, which is known to be required for early Clb5 degradation and is specific for yeast Cdc20[15]. Thus, activator specificity depends on the D box in some cases while in others is provided by a second degron. In the cell, early Clb5 degradation is also promoted by the associated

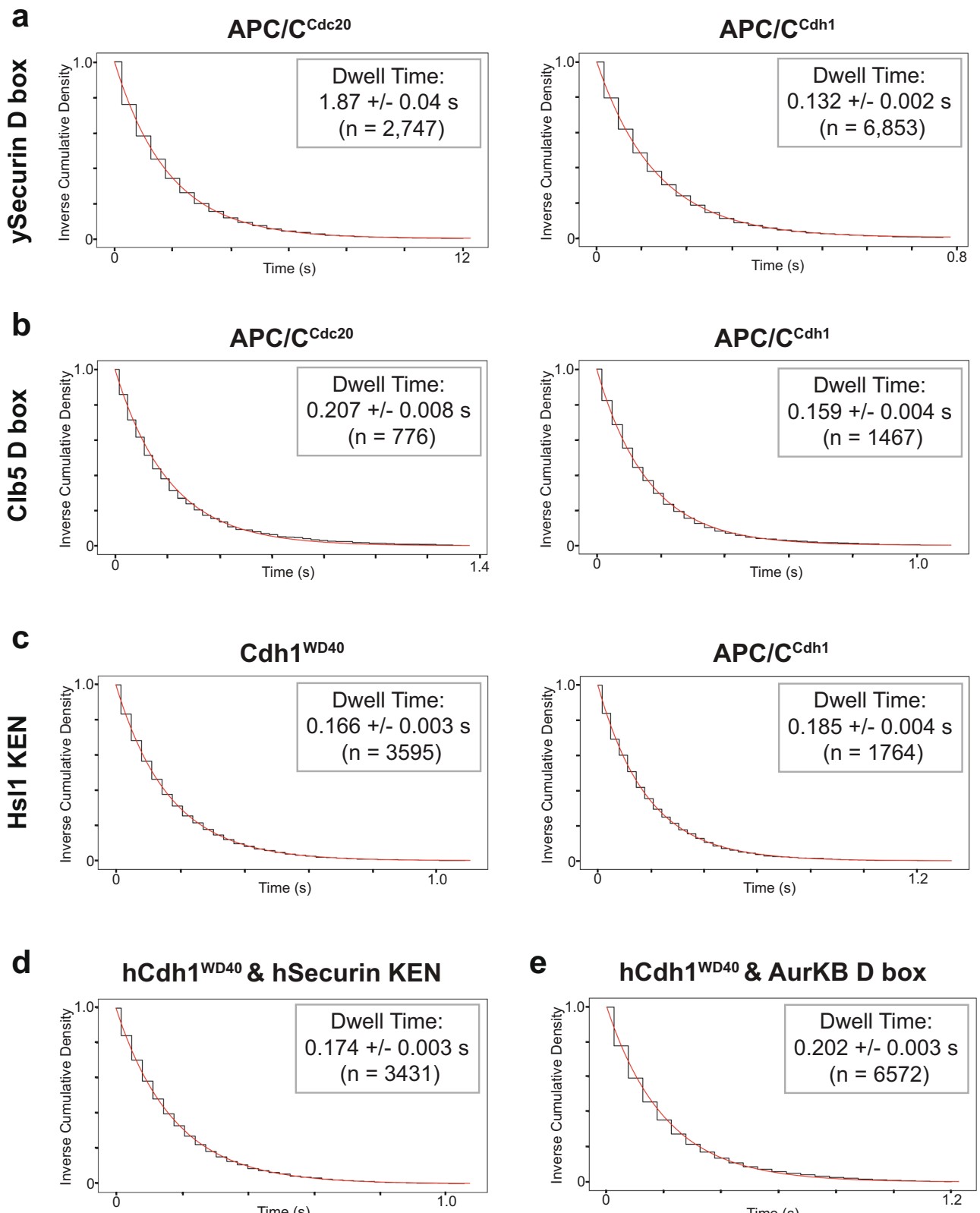

**Fig. 5 Analysis of degrons from multiple substrates. a–c** Dwell time distributions from SMOR analysis of representative movies with Cy5-labeled ySecurin D box peptide (**a** 1 nM), Clb5 D box peptide (**b** 1 nM), and Hsl1 KEN peptide (**c** 1 nM) binding to the indicated GFP-tagged proteins. **d, e** Analysis of hSecurin KEN box peptide (**d** 500 pM) or AurKB D box peptide (**e** 1 nM) binding to GFP-tagged human Cdh1 WD40 domain. Insets indicate mean dwell time ± SE (*n* indicates number of selected peaks). Results are representative of two independent experiments (Supplementary Table 1).

**Table 2 Dwell times for substrate interactions with human activators.**

|  | hCdh1$^{WD40}$ | hCdc20$^{WD40}$ |
|---|---|---|
| Hsl1 D box | 4.52 s | 1.08 s |
| Hsl1 D box mutant | No binding | No binding |
| ySecurin D box | SF | SF |
| Clb5 D box | SF | 0.054 s$^a$ |
| Hsl1 KEN | 1.48 s | SF |
| ySecurin KEN | 0.869 s | SF |
| Acm1 ABBA | No binding | No binding |
| AurKB D box | 0.202 s | SF |
| hSecurin KEN | 0.174 s | No binding |

Dwell times are representative of two independent replicates, listed in Supplementary Table 1.
SF single frame.
$^a$Calculated dwell time is less than three times the frame rate and is, therefore, less reliable.

Cks1 subunit, which is thought to interact with phosphorylated sites on the APC/C[15].

Interestingly, the KEN box of securin is specific for APC/C$^{Cdh1}$ and does not bind APC/C$^{Cdc20}$. Residues outside the KEN motif must influence binding to different features on the two activators. As securin is known to be preferred by Cdc20 in vivo and in ubiquitylation assays, the preference of its KEN box for Cdh1 must not overcome the stronger preference of its D box for Cdc20.

We did not observe the binding of KEN peptides to Cdc20 using yeast or human Cdc20 with yeast or human KEN peptides, respectively (Tables 1 and 2). The only case in which we observed an interaction was single-frame binding of yeast KEN peptides to human Cdc20 (Table 2). The KEN degron was originally identified as a motif targeted by Cdh1[12], and there is evidence to support Cdh1 specificity of the KEN box in some substrates. In Cyclin A2, for example, the KEN box is more important for ubiquitylation by APC/C$^{Cdh1}$ than by APC/C$^{Cdc20}$, whereas D boxes are more important for ubiquitylation by APC/C$^{Cdc20}$ [18]. KEN degrons might increase the Cdh1 affinity of substrates with Cdc20-specific D boxes, ensuring that these substrates continue to be unstable in late mitosis and G1.

In contrast to our evidence that the KEN box binds poorly, if at all, to Cdc20, there is structural evidence for Cdc20 binding to KEN degrons from BubR1 and Cyclin A2[18,38]. In these structures, the KEN box is part of a protein containing additional degrons, and it seems likely that KEN binding to a low-affinity site on Cdc20 is driven in these cases by the high local concentration provided by a multivalent ligand.

Our studies also provide a quantitative understanding of the contributions of activator and Apc10 to the composite D-box binding site. The affinity of the Hsl1 D box for APC/C$^{Cdh1}$ is 100-fold higher than that with Cdh1 alone or with an APC/C carrying an Apc10 mutation, showing the dramatic impact of Apc10 on D box affinity. D-box binding can be considered as a bivalent interaction, in which the N-terminal RxxL segment of the degron binds with moderate affinity ($K_D = 3.5$ μM) to specific sites on the activator surface, while poorly-conserved sequences at the C-terminal end of the D box interact weakly with Apc10. The latter interaction is not well understood at the structural level and is clearly very low affinity. We observed no binding of D box peptides to APC/C$^{apo}$ or Apc10, and the only reported evidence for direct binding comes from NMR analysis of Apc10-D box interactions at high (5 mM) concentrations of Hsl1 D box peptide[39]. Nevertheless, this low-affinity Apc10 interaction cooperates effectively with the moderate-affinity activator interaction to generate high-affinity bivalent binding of the D box to APC/C$^{Cdh1}$.

The D-box binding pocket is well-conserved in Cdc20 and Cdh1, but our results suggest that the two activators present the D box to Apc10 in different ways. Although Apc10 boosted affinity for the Hsl1 D box by 100-fold in the case of APC/C$^{Cdh1}$, it seemed to provide only a ~3-fold increase in binding to APC/C$^{Cdc20}$. Similarly, binding to APC/C$^{Cdc20}$ of Hsl1 containing both D and KEN boxes is only slightly reduced by mutation of Apc10. Perhaps the Cdc20-D box complex is oriented in a way that results in a low-affinity interaction with Apc10[18,40].

Most if not all APC/C substrates contain multiple degrons, and our studies document the high affinity that results from the multivalent binding of D and KEN boxes of Hsl1. When a moderate affinity D box ($K_D = 3.5$ μM) exists on the same protein as a moderate affinity KEN box ($K_D = 14$ μM), the result is an Hsl1 dwell time of over 300 s—suggesting a dissociation constant of ~4 nM. The effects of multivalency are further enhanced by the allosteric enhancement of each degron's binding when the other is bound (Fig. 1a, c).

The Hsl1 dwell time of 5 min with APC/C$^{Cdh1}$ is a very long time in the life of a yeast cell, which divides every 90 min. This raises the possibility that Hsl1 does not dissociate spontaneously from the APC/C but is extracted from the APC/C by the proteasome. However, Hsl1 may be an unusual case, as suggested by the unusually high affinity of its degrons. The degrons of securin and Clb5 have lower affinities than those of Hsl1, and these substrates are therefore likely to bind with lower affinity. Unfortunately, we were unable to measure the binding of securin and other substrates due to their tendency to aggregate and create excessive background fluorescence in our assay.

In sum, our results reveal that the affinity and specificity of the APC/C-substrate interaction can be influenced by a remarkable array of factors. Key factors include the specificity and affinity of individual degrons for the activator and Apc10, as well as the presence of multiple degrons on a substrate. Numerous other factors are also likely to be important, such as the number and positioning of lysines for modification, the distance between degrons, and the orientation of the D box at its bivalent binding site[7,16,18,40].

The timing of substrate ubiquitylation and destruction is important for robust control of cell cycle events. Our past work suggests that substrate affinity is a key determinant of the timing of substrate degradation[15,22], and there might be some contribution from competition among substrates[23]. A full understanding of the ordering of substrate degradation will require more extensive studies of the concentrations and affinities of substrates and the APC/C inside the cell.

Using tools developed by single-molecule biophysics, the SMOR assay provides a straightforward approach for biochemists to study macromolecular interactions that are difficult to study by conventional methods. Although our experiments were performed with purified components, we suspect that the SMOR assay will also be effective for studying binding proteins that are purified directly on the glass[31]. The continued development of single-molecule approaches promises to open many new avenues in the study of biological and therapeutic interactions.

## Methods

**Yeast APC/C purification.** Yeast strains were derivatives of W303 and are listed in Supplementary Table 2. For APC/C purification, we used a strain carrying Cdc16-TAP and lacking Cdh1 (DOM1126); in most experiments, the strain also carried Apc1-GFP (NHY13). Yeast were grown in YPD media to $OD_{600} = 0.8$, collected and flash-frozen. Cells were lysed by bead beating in lysis buffer A (50 mM HEPES pH 7.5, 150 mM KCl, 10% glycerol, 0.2% Triton X-100, 63 μM B-glycerophosphate, 48 μM sodium fluoride, 1 μg ml$^{-1}$ pepstatin A, 1 μg ml$^{-1}$ leupeptin, 1 μg ml$^{-1}$ aprotinin, 1 mM PMSF, 1 mM DTT, 0.5 mM EDTA) and APC/C was purified using magnetic IgG beads. The beads were washed using wash buffer A (20 mM HEPES pH 7.5, 150 mM KCl, 10% Glycerol, 0.05% Triton X-100). After incubation

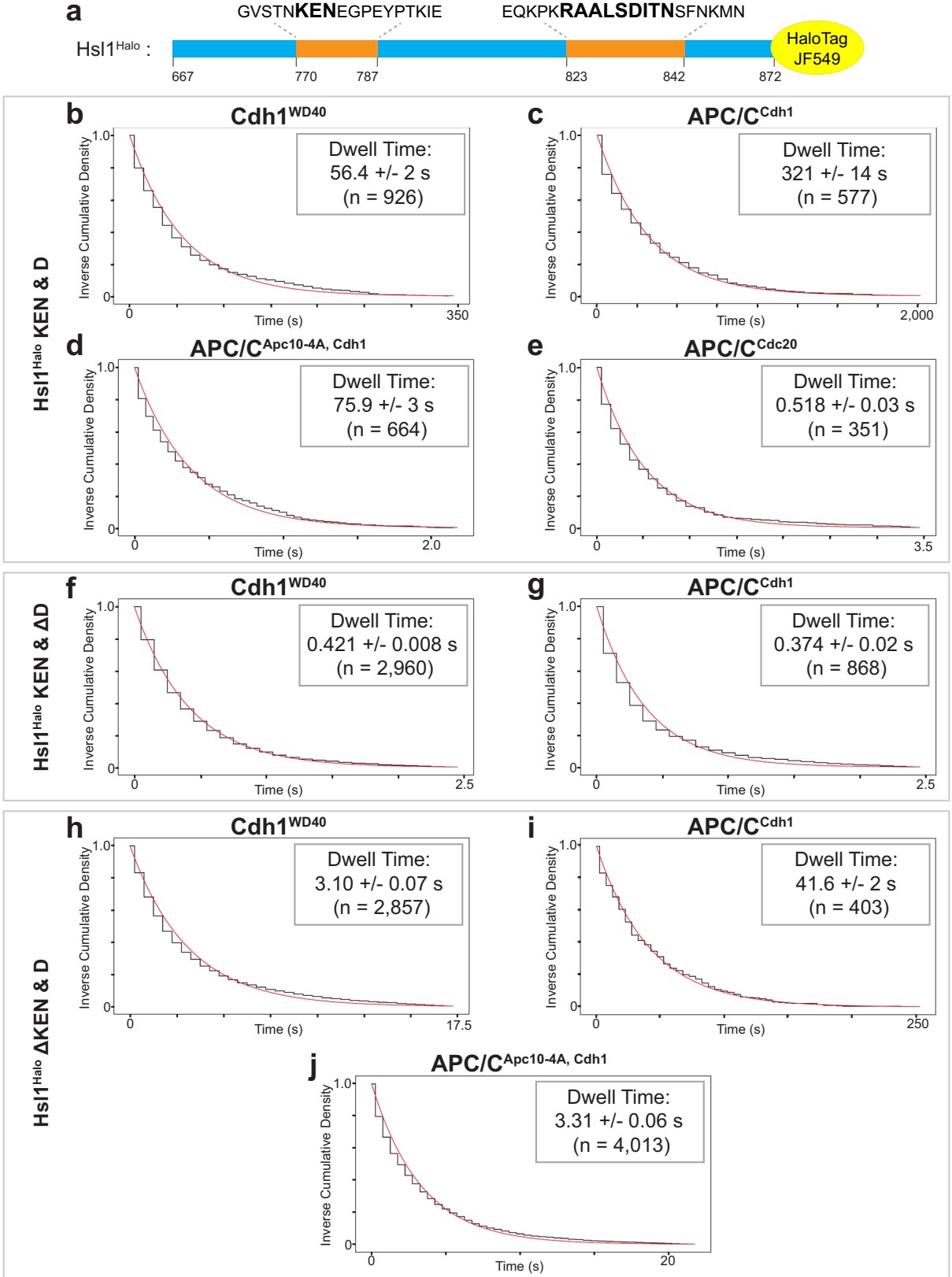

**Fig. 6 Double degron substrate interactions. a** Hsl1$^{Halo}$ fragment used in these experiments, showing sequences of KEN and D boxes. **b–e** Dwell time distributions from SMOR analysis of representative movies with Hsl1$^{Halo}$ carrying wild-type degrons and the indicated GFP-tagged binding proteins: **b** Cdh1$^{WD40}$; **c** APC/C$^{Cdh1}$; **d** APC/C$^{Cdh1}$ with Apc10-4A mutations; **e** APC/C$^{Cdc20}$. **f–g** Analysis of Hsl1$^{Halo}$ with mutant D box, binding to: **f** Cdh1$^{WD40}$; **g** APC/C$^{Cdh1}$. **h–j** Analysis of Hsl1$^{Halo}$ with mutant KEN, binding to: **h** Cdh1$^{WD40}$; **i** APC/C$^{Cdh1}$; **j** APC/C$^{Cdh1}$ with Apc10-4A mutations. Insets indicate mean dwell time ± SE (n indicates number of selected peaks). Results are representative of two independent experiments (Supplementary Table 1).

with purified Cdh1 or Cdc20, the APC/C was cleaved off the beads with TEV protease in wash buffer A with 0.05% Tween-20 (for single-molecule studies) or 0.05% Triton X-100 (for ensemble assays such as ubiquitylation) and used immediately.

**Activator purification**. Activators were cloned into the pFastBac HT A vector, with an N-terminal 2xStrep-Tag II. For some experiments, we constructed vectors for expression of the WD40 domains of yeast Cdh1 (aa 241–550; pNH144), human Cdh1 (aa 165–484; pNH164), or human Cdc20 (aa 162–484; pNH188). For SMOR, a C-terminal GFP tag was added. Bacmid plasmids are listed in Supplementary Table 3. Plasmids were transformed into DH10Bac cells, and purified Bacmid was used to transfect Sf9 cells to generate P1 baculovirus, which was used to generate P2 virus. SF9 cells were infected with the P2 virus for 48 h. Flash-frozen pellets were lysed by sonication or high-pressure homogenization in lysis buffer B (50 mM Tris-HCl pH 8, 150 mM KCl, 0.5 mM EDTA, 1 mM DTT, 10% Glycerol, EDTA-free protease inhibitor tablet, 1 µg ml$^{-1}$ pepstatin A). After the lysate was applied to the StrepTrap column, the column was washed with wash buffer B (lysis buffer B lacking protease inhibitors) and eluted in the same buffer containing 2.5 mM desthiobiotin.

**Polarization anisotropy**. Fluorescent degron peptides carried a C-terminal Cy5 label (CPC Scientific) and are listed in Supplementary Table 4. For anisotropy experiments, 10 nM fluorescent peptide was mixed with various concentrations of purified Cdh1 WD40 in wash buffer B at room temperature for 1 min, which we determined was sufficient for the binding reaction to reach equilibrium. Fluorescence was measured on a K2 Multifrequency Fluorometer at 25 °C. All Cy5-labeled peptides were excited with polarized light at 635 nm and emission was detected using a 700/75 nm bandpass filter (ET series, Chroma). A competition experiment was conducted with an unlabeled Hsl1 D box peptide to confirm that the Cy5 dye did not bind the Cdh1 WD40. Data were fitted to one-site equilibrium binding using GraphPad Prism version 8.4 to determine $K_D$. Results were the same when peptide binding was measured in the buffer used in SMOR assays (buffer C, below).

**SMOR assay surface preparation and protein immobilization**. For all SMOR experiments, 24 × 50 mm high precision glass coverslips (Bioscience Tools) and drilled microscope slides were passivated with a combination of PEG and PEG-biotin (coverslips) or PEG only (slides) following a previously published protocol for SiMPull[31]. Reaction chambers (~20 µl) were created using double-sided tape and epoxy. Protein was immobilized on the surface with 0.2 mg ml$^{-1}$ NeutrAvidin in buffer C (20 mM HEPES pH 7.5, 150 mM KCl, 10% glycerol, 0.05% Tween-20, 1 mM DTT). Excess NeutrAvidin was washed away and incubated with either biotin-conjugated mouse monoclonal anti-Strep-Tag II antibody (LSBio, Cat. No. LS-C203632-100) or biotin-conjugated rabbit polyclonal anti-GFP antibody (Rockland, Cat. No. 600-406-215) diluted in buffer C + BSA (0.1 mg ml$^{-1}$; Molecular Biology Grade Bovine Serum Albumin). Activator WD40 and activated APC/C (typically 50 µl containing about 65 ng APC/C, of which only a small fraction is immobilized on the glass) were immobilized using anti-Strep-Tag II, while APC/C alone was immobilized with anti-GFP. SMOR results were similar when performed in the buffer (wash buffer B) used in polarization anisotropy.

**Kinetics experiments with SMOR assay**. To capture dynamic protein–protein interactions, dye-labeled substrate diluted in buffer C + BSA was added to the chamber containing immobilized proteins. Interactions were imaged by TIRF microscopy as described below. For optimal signal-to-noise ratio, peptide concentration was no greater than 1 nM. Protein substrates included Hsl1 aa 667–872 and a C-terminal HaloTag followed by a TAP tag, and were produced by translation in rabbit reticulocyte lysates using TnT Quick Coupled Transcription/Translation System (Promega) (see Supplementary Table 5 for plasmids). Protein was purified with magnetic IgG beads and labeled on the bead with JF549 or JF646 dye so that unbound dye could be washed away. Purified dye-labeled protein was then cleaved from the beads using TEV protease. Before adding substrate to the reaction chamber, oxygen scavenging reagents were added (10 nM proto-catechuate-3,4-dioxygenase, 2.5 mM protocatechuic acid, and 1 mM Trolox)[41]. A detailed description of the SMOR analysis pipeline is found in Supplementary Note 1.

**Single-molecule TIRF microscopy**. Microscopy was performed with a Nikon Eclipse TE2000-E with Perfect Focus with a 100 × 1.49na oil, Apo TIRF DIC N2, 0.13-0.2, WD0.12 objective. Movies were recorded using the Andor iXon DU-897E-CSO. For visualizing GFP-tagged protein, a 491 nm laser was used with an ET525/50 (Chroma) filter for emission. For visualizing JF549-labeled substrate, a 561 nm laser was used with an ET595/50 (Chroma) filter for emission. For visualizing all Cy5-labeled peptides and JF646-labeled substrates, a 640 nm laser was used with an ET685/70 filter (Chroma) for emission. Initially, for each substrate tested, we acquired movies at multiple intervals to deduce the optimal interval to decrease bleaching. µManager version 2.0 was used to control the microscope and record time-lapse movies.

**Ubiquitylation assay**. APC/C ubiquitylation assays were performed as described previously[24]. In short, APC/C was purified from yeast as described above using magnetic IgG beads and activated with a purified Cdh1 or Cdc20 activator. For all APC/C ubiquitylation assays, substrates were produced by in vitro translation with $^{35}$S-Methionine (see Supplementary Table 5 for plasmids). Substrates were purified using magnetic IgG beads. E1 and E2 (Ubc4) were expressed in *E. coli* and purified as described previously[25,42]. E2 was charged at 37 °C for 30 min (reaction contained 0.2 mg ml$^{-1}$ Uba1, 2 mg ml$^{-1}$ Ubc4, 2 mg ml$^{-1}$ methylated ubiquitin from Boston Biochem #Y-501, and 1 mM ATP). Charged E2 was added to a reaction containing activated APC/C and substrate. Reaction products were analyzed by SDS-PAGE and Phosphorimaging on a Typhoon 9400 Imager.

**Reporting summary**. Further information on research design is available in the Nature Research Reporting Summary linked to this article.

## Data availability

All video and single-molecule analysis data generated in this study are available on a public repository (https://zenodo.org/record/5726046). Source data are provided with this paper.

## Code availability

Code for analysis of video data is available in a Github repository (https://github.com/jmsung/smor-analysis) and an explanation is provided in Supplementary Note 1.

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

## Acknowledgements
We thank Nico Stuurman, DeLaine Larson, and Kurt Thorn for their expert advice on microscopy, Ron Vale for advice and support, Luke Lavis and Margaret Jefferies (HHMI) for generously sharing fluorescent dyes, Diego Garrido Ruiz and Matt Jacobson for discussions of activator structure, and Sy Redding, Lucy D. Brennan, and Stephanie L. Johnson for their invaluable advice about single-molecule analysis. We would also like to thank members of the Morgan lab for valuable discussions and Hayk R. Mangasarian for his patience. This work was supported by a Graduate Research Fellowship from the National Science Foundation (N.H.), the Paul and Daisy Soros Fellowship for New Americans (N.H.), the UCSF Discovery Fellowship (N.H.), and the National Institute of General Medical Sciences (R35-GM118053, to D.O.M.).

## Author contributions
N.H. and D.O.M. conceived the single-molecule methods and experiments. N.H. designed and performed the experiments. J.S. created the image analysis code. A.J. helped develop single-molecule methods. N.H. and D.O.M. wrote the paper with assistance from all authors.

## Competing interests
The authors declare no competing interests.
