## [Peer Review File · Nature Communications]

Single-molecule analysis of specificity and multivalency in binding of short linear substrate motifs to the APC/CReviewers' Comments:

Reviewer #1:

Remarks to the Author:

The manuscript by Morgan and colleagues explores the affinity of APC/C-activator complexes for APC/C degrons that are encoded as SLiMs, specifically the D and KEN boxes and ABBA motif. To do this they developed a novel single molecule off-rate (SMOR) assay based on TIRF microscopy and dwell times. This allowed the authors to address important long-standing questions in APC/C function and mechanism: namely the role of the two activator subunits Cdc20 and Cdh1 in determining APC/C specificity for different substrates, and the roles of specific degrons, that account for cell cycle-dependent changes in substrate ubiquitination.

This study is carefully performed. The question of how to measure affinities of proteins and other ligands to large, low abundant complexes is of critical interest and importance. Thus, the authors have developed an important and valuable tool that will be useful for further examination of APC/C affinities for its substrates and regulators, and be applicable to understanding other large molecular complexes and machines. The application of this method to APC/C substrates also answers and explains key questions in the APC/C and cell cycle fields. For example, the authors have shown that the KEN box confers virtually no binding affinity to APC/C-Cdc20 complexes, whereas the affinity of APC/C-Cdh1-peptide complexes are enhanced with the KEN motif. Further the authors confirm that Apc10 enhances binding for D box containing motifs in the context of APC/C-Cdh1 by 100-fold. However, surprisingly, using the Apc10 mutant, the authors show that Apc10 does not contribute strongly to enhancing D-box binding to APC/C-Cdc20. Whereas the Hsl1 D box has a higher affinity for APC/C-Cdh1, compared with APC/C-Cdc20, the authors were able to show that the ysecurin D box binds APC/C-Cdc20 more strongly than APC/C-Cdh1, consistent with securin being ubiquitinated by APC/C-Cdc20 in vivo. An exciting result was the authors finding that although the KEN box of Hsl1 alone has quite low affinity for APC/C-Cdh1 (and Cdh1-WD40), combining the KEN box with D box greatly increases the affinity of the Hsl1 peptide by 100 to 200-fold, indicating how these two degron cooperative to substantially enhance affinity.

The manuscript is well written, including the novel methodology, and the figures are logical and clear.

I recommend publication of this manuscript in Nature Communications, subject to the authors addressing questions and comments below:

1. Fig.1 The authors report that there is positive allosteric cooperativity between the D box and KEN box degron sites based on the enhanced affinity of Cdh1-WD40 for the D box peptide in the presence of a KEN box peptide (Fig. 1d).
 - a. However, to this reviewer some uncertainty exists on this point. First, at least for KEN box enhancement by D box of 2-fold, this seems close to the experimental error, since in Fig. 1c, the K_d of the Hsl1 KEN is 12 μ M but in Fig. 1d it is 6 μ M. What is the SD for the K_d values?
 - b. To support the idea of cooperativity acting via the WD40 domain, the authors could consider mutating either the D box or KEN box site on Cdh1-WD40 and testing again if cooperativity exists.
 - c. The SD values of K_d s should be indicated.
2. Fig. 1b. List human peptide sequences used in this study
3. APC/C-Cdh1 interaction stability and life-time. The SMOR assay is dependent on a stable APC/C-Cdh1 interaction, longer than the dwell times of APC/C-activator-substrate complexes. Can the authors measure the dwell time of APC/C bound to activator?
4. The authors measured ysecurin D box, and human KEN box for APC/C-activators, but not hsecurin D box. Is there a reason for this omission?
5. Extended Data Fig. 6: Please include K_d s.
6. End of page 11. The explanation isn't clear. Where does the 10-fold difference between the range of intensities found in the max and min intensity projections (Ext Data Fig 4a) come from? Seems more

like 1

Reviewer #2:

Remarks to the Author:

Nairi Hartooni and coauthors have reported single molecule measurements to quantify how small peptides bind to a protein complex. Here, the peptides represent the short linear motifs (SLiMs) in the substrates (D box and KEN box) of a protein complex called Anaphase-Promoting Complex/Cyclosome (APC/C). This is a significant problem.

I have a few significant concerns:

1. The authors have used SLiM peptides labeled with a single color and performed single color measurements, which may not be the most reliable way to measure transient binding events. Instead, the authors can think of single molecule FRET measurements. One can label a component in the immobilized complex (say acceptor dye) and then flow labeled peptides with the corresponding donor dye. Every time there is a binding event, one would measure the unmistakable FRET signal. If photobleaching of dyes is controlled using an oxygen scavenger system or more photostable dyes are used, FRET may reveal dynamics and dwell time to quantify K_{off} rate.
2. Example traces in the paper appear to be heavily smoothed. Also, single molecule measurements are notorious for having artifacts. All kinds of effects can be observed even with an empty slide due to impurities and dyes.
3. Single-step photobleaching of dyes can potentially be leveraged if smFRET is difficult to perform with this system.

Without these technical controls, I do not feel confident that scientific conclusions can be drawn. Therefore, I have refrained from commenting on the analyses.

Response to Reviewers
Hartooni et al. [Nature Communications NCOMMS-21-39171-T]

We thank the reviewers for their thoughtful comments and suggestions. Our responses to these comments are provided below in blue font. In the text of the paper, major changes are highlighted in gray.

Reviewer #1:

The manuscript by Morgan and colleagues explores the affinity of APC/C-activator complexes for APC/C degrons that are encoded as SLiMs, specifically the D and KEN boxes and ABBA motif. To do this they developed a novel single molecule off-rate (SMOR) assay based on TIRF microscopy and dwell times. This allowed the authors to address important long-standing questions in APC/C function and mechanism: namely the role of the two activator subunits Cdc20 and Cdh1 in determining APC/C specificity for different substrates, and the roles of specific degrons, that account for cell cycle-dependent changes in substrate ubiquitination.

This study is carefully performed. The question of how to measure affinities of proteins and other ligands to large, low abundant complexes is of critical interest and importance. Thus, the authors have developed an important and valuable tool that will be useful for further examination of APC/C affinities for its substrates and regulators, and be applicable to understanding other large molecular complexes and machines. The application of this method to APC/C substrates also answers and explains key questions in the APC/C and cell cycle fields. For example, the authors have shown that the KEN box confers virtually no binding affinity to APC/C-Cdc20 complexes, whereas the affinity of APC/C-Cdh1-peptide complexes are enhanced with the KEN motif. Further the authors confirm that Apc10 enhances binding for D box containing motifs in the context of APC/C-Cdh1 by 100-fold. However, surprisingly, using the Apc10 mutant, the authors show that Apc10 does not contribute strongly to enhancing D-box binding to APC/C-Cdc20. Whereas the Hsl1 D box has a higher affinity for APC/C-Cdh1, compared with APC/C-Cdc20, the authors were able to show that the ysecurin D box binds APC/C-Cdc20 more strongly than APC/C-Cdh1, consistent with securin being ubiquitinated by APC/C-Cdc20 in vivo. An exciting result was the authors finding that although the KEN box of Hsl1 alone has quite low affinity for APC/C-Cdh1 (and Cdh1-WD40), combining the KEN box with D box greatly increases the affinity of the Hsl1 peptide by 100 to 200-fold, indicating how these two degron cooperative to substantially enhance affinity.

The manuscript is well written, including the novel methodology, and the figures are logical and clear.

I recommend publication of this manuscript in Nature Communications, subject to the authors addressing questions and comments below:

1. Fig.1 The authors report that there is positive allosteric cooperativity between the D box and KEN box degron sites based on the enhanced affinity of Cdh1-WD40 for the D box peptide in the presence of a KEN box peptide (Fig. 1d).
 - a. However, to this reviewer some uncertainty exists on this point. First, at least for KEN box enhancement by D box of 2-fold, this seems close to the experimental error, since in Fig. 1c, the K_d of the Hsl1 KEN is 12 μ M but in Fig. 1d it is 6 μ M. What is the SD for the K_d values?

We thank the reviewer for the positive comments and careful consideration of our data. These are excellent points, and we agree that the original version of Figure 1 did not provide sufficient detail about error in peptide binding results. We have now redone this figure to include more detail about these issues.

1. Error in KD values. The binding curves in Figure 1 are single representative experiments of 2 or 3 experimental replicates. In a single replicate like those shown, all data points (each the average of 10 reads in the fluorimeter) are collected from parallel binding reactions done at the same time with the same Cdh1 preparation. The GraphPad Prism software determines an ideal fit for a simple binding interaction, generating a KD value. Although an estimate of SE can be calculated by Prism for a single experiment, SE is best determined by obtaining the mean KD in independent experiments. This is what we have now added to Figure 1. The table of peptides now provides mean KD +/- SE in 2 or 3 experimental replicates. As before, curves from one replicate are provided in the other panels to illustrate unprocessed representative results.

2. Variation in KEN binding affinity. The reviewer is correct in pointing out the variation in the KD values for KEN peptide binding (particularly the yeast securin KEN). This variation is a result of technical limitations: we cannot achieve high enough Cdh1 concentrations to reach KEN binding saturation (ideally 10-fold over KD). Thus, KEN-binding experiments performed on different days with different Cdh1 preparations can produce slightly different binding curves and KD values that are less reliable than those we obtain with the higher-affinity D box peptide. Variation in KEN KD between independent experiments is therefore expected under these non-saturating conditions. We have rewritten the Results text to state that KD values for KEN peptides are potentially inaccurate because of our inability to reach binding saturation.

3. Allosteric effects of degron binding. The effects of dark peptides on fluorescent degron binding affinity (now Fig. 1c) were measured at the same time with the same Cdh1 preparation, and thus inter-experiment variability for the KEN affinity is not a significant issue – particularly because the dark D box causes KEN affinity to rise to a much more reproducible high affinity (low KD). We are therefore confident in our conclusion that dark peptides cause a 3-4-fold improvement in affinities for both degrons.

b. To support the idea of cooperativity acting via the WD40 domain, the authors could consider mutating either the D box or KEN box site on Cdh1-WD40 and testing again if cooperativity exists.

This is an excellent idea. Several years ago, however, we found that point mutations in the fragile WD40 domain often have deleterious effects on protein stability. For example, we mutated the Cdh1 residues involved in D box binding and found that the resulting protein was sufficient for some purposes but had defects in APC/C binding and stability. We are therefore doubtful about using such mutants to assess the subtle allosteric effects we describe in our current work. However, we did pursue an alternative approach that addresses the reviewer's concern: we measured the effects of dark peptides lacking the key consensus residues of the degron sequence. These results are included in the new Figure 1c. In both cases, these mutant peptides had no effect on the affinity for the other degron, confirming that it is binding to degron-binding sites that drives the allosteric effect.

c. The SD values of Kds should be indicated.

The table in Figure 1a now provides Mean +/- SE for the KD values described in the figure. We have also added Prism-calculated SE values to the KD values for Hsl1 D box binding to human

Cdh1 in Supplementary Figure 6.

2. Fig. 1b. List human peptide sequences used in this study.

Now added to Supplementary Figure 6.

3. APC/C-Cdh1 interaction stability and life-time. The SMOR assay is dependent on a stable APC/C-Cdh1 interaction, longer than the dwell times of APC/C-activator-substrate complexes. Can the authors measure the dwell time of APC/C bound to activator?

Excellent point. We neglected to mention that we studied activator dissociation from the APC/C in our previous work (Mizrak & Morgan, 2019), and our results showed that a purified APC/C-activator complex does not dissociate over a 30-minute time course. We therefore believe that activator affinity is extremely high in vitro (but not in vivo, where other factors promote dissociation). We have added a small new paragraph on this point.

4. The authors measured ysecurin D box, and human KEN box for APC/C-activators, but not hsecurin D box. Is there a reason for this omission?

We tried the human securin D box peptide but found that it exhibited excessive background binding, resulting in bright multimers on glass coated with antibody only, as seen in the image below. Accurate measurement of its binding to activator or APC/C was therefore not possible. Given that many readers might wonder about this omission, we have added a sentence mentioning the poor behavior of this peptide.

5. Extended Data Fig. 6: Please include Kds.

These have now been added for the Hsl1 D box. Signals with the other peptides in this figure are not sufficient to allow calculation of a reliable KD value.

6. End of page 11. The explanation isn't clear. Where does the 10-fold difference between the range of intensities found in the max and min intensity projections (Ext Data Fig 4a) come from? Seems more like 1

Note that the images in Supp Fig 4a are not at the same intensity, as illustrated by the 10-fold difference in the intensity scales to the right of each image. Thus, the dots in the max intensity projection on the right are actually much brighter than the few dots seen in the minimum intensity projection on the left. Is this the source of the concern?

Reviewer #2:

Nairi Hartooni and coauthors have reported single molecule measurements to quantify how

small peptides bind to a protein complex. Here, the peptides represent the short linear motifs (SLiMs) in the substrates (D box and KEN box) of a protein complex called Anaphase-Promoting Complex/Cyclosome (APC/C). This is a significant problem.

I have a few significant concerns:

1. The authors have used SLiM peptides labeled with a single color and performed single color measurements, which may not be the most reliable way to measure transient binding events. Instead, the authors can think of single molecule FRET measurements. One can label a component in the immobilized complex (say acceptor dye) and then flow labeled peptides with the corresponding donor dye. Every time there is a binding event, one would measure the unmistakable FRET signal. If photobleaching of dyes is controlled using an oxygen scavenger system or more photostable dyes are used, FRET may reveal dynamics and dwell time to quantify Koff rate.

We agree that a smFRET approach could be useful; indeed, when we first began developing these methods 4-5 years ago we considered this option. However, we decided against it for the following reasons:

a. Most importantly, our goal was to develop a straightforward, robust method that can be adapted quickly by non-biophysicists for measuring any protein-protein interaction, without the need for the time-consuming trial and error experiments that are generally needed for the positioning of fluorescent dyes in FRET-based methods.

b. Our approach is not biased by preconceptions about the location of binding sites on large macromolecular complexes like the APC/C. Unlike FRET-based binding methods, our method does not require previous knowledge of the protein structure for placement of the dyes. Furthermore, a FRET-based approach will fail to detect binding at unexpected locations far from the acceptor dye.

c. Even with our detailed knowledge of APC/C and activator structure, it is not entirely clear how or where we would place acceptor dyes that would provide a useful readout for measuring binding of different degrons at different sites. Exploring a variety of possible dye locations in APC/C subunits would be very time-consuming.

d. Given the extensive control experiments and computational methods we describe in the paper, we are confident that measurements of a single color fluorescent spot provides a highly reliable way to identify binding events and dissociation rate constants.

e. As described in the Results section, we do use oxygen scavengers to reduce photobleaching, which does not occur at a significant rate on the time scale of the binding events we are measuring.

2. Example traces in the paper appear to be heavily smoothed. Also, single molecule measurements are notorious for having artifacts. All kinds of effects can be observed even with an empty slide due to impurities and dyes.

Yes, the traces in Fig. 3d & e have been smoothed, which is an integral part of the processing we perform for analysis of each movie. Below are two representative raw traces, in which the high signal-to-noise ratio is apparent. On the left is a measurement of Hsl1 D box binding to APC/C-Cdh1 (1000 frames at 100 ms frame rate, continuous imaging); on the right is Hsl1-Halo binding to APC/C-Cdh1 (1000 frames at 32 ms frame rate, continuous imaging). These

particular traces were obtained in a control experiment under photobleaching conditions: bleaching occurred in a single step, confirming that the binding event involved a single fluorescent ligand (discussed below).

We agree that all single molecule studies can have artifacts due to impurities, background noise, etc. Indeed, we dedicated considerable effort to refining our glass preparation methods, which were originally developed by one of us (Ankur Jain) as a graduate student in the lab of Taekjip Ha (ref 31 in the paper). Numerous control experiments (Figures 2, 3; Supplementary Figures 3, 7, and 8) demonstrate the remarkably low background in our experiments. In Figure 2b, for example, there is essentially no APC/C binding to the glass in the absence of antibody, and in Supplementary Figure 3 it is apparent that there are few background spots on glass coated only with antibody. Most importantly, these background spots often represent multimers that are discarded during the analysis by the methods we use to isolate only those spots that represent single molecules (Fig. 3c). We only studied fluorescent ligands that met our stringent criteria for specific binding (at least 10-fold higher specific binding than nonspecific binding). For example, this is why we did not pursue studies of the human securin D box, as mentioned above.

3. Single-step photobleaching of dyes can potentially be leveraged if smFRET is difficult to perform with this system.

As described in the Results, our analysis pipeline includes a step designed specifically to discard fluorescent spots that are unlikely to represent single molecules. The approach (Fig. 3c) is to plot a fluorescence intensity histogram. The distribution is unimodal, indicating that there is one type of fluorescent species in each spot – likely a single fluorescent molecule. The analysis then discards spots that are 3 standard deviations outside the median. As mentioned above, we also performed control experiments with all ligands under bleaching conditions, showing that bleaching occurs in a single step.

Without these technical controls, I do not feel confident that scientific conclusions can be drawn. Therefore, I have refrained from commenting on the analyses.

Reviewers' Comments:

Reviewer #1:

Remarks to the Author:

The authors have satisfactorily addressed my questions.

I was pleased to see the new data in point 1b.

The manuscript is of interest and importance that should be published in Nature Communications.

Reviewer #2:

Remarks to the Author:

I am satisfied with the revision and recommend publications. However, I would prefer that the authors comment on two relevant questions, if possible, because affinity and stoichiometry are relevant information that can be obtained from the data:

(1) Since the APC/C-activator complex does not dissociate even after 30 min, the affinity must be extraordinarily strong (pM or higher) as the authors have noted. Although there is a new paragraph in the revision, a little more quantitative approach might be useful. I expect that the distribution of dwell times to be exponential in the absence of any cooperativity and 1:1 stoichiometry. The decay rate of an exponential fit can then be used to approximate the binding strength using Eyring equation.

(2) Do the authors only observe single-step photobleaching? Counting photobleaching steps may inform or confirm stoichiometry after controlling for the labelling efficiency.

Response to Reviewers
Hartooni et al. [Nature Communications NCOMMS-21-39171A]

Reviewer #1 (Remarks to the Author):

The authors have satisfactorily addressed my questions.

I was pleased to see the new data in point 1b.

The manuscript is of interest and importance that should be published in Nature Communications.

We are grateful to the reviewer for providing a number of excellent suggestions, resulting in a greatly improved paper.

Reviewer #2 (Remarks to the Author):

I am satisfied with the revision and recommend publications. However, I would prefer that the authors comment on two relevant questions, if possible, because affinity and stoichiometry are relevant information that can be obtained from the data:

(1) Since the APC/C-activator complex does not dissociate even after 30 min, the affinity must be extraordinarily strong (pM or higher) as the authors have noted. Although there is a new paragraph in the revision, a little more quantitative approach might be useful. I expect that the distribution of dwell times to be exponential in the absence of any cooperativity and 1:1 stoichiometry. The decay rate of an exponential fit can then be used to approximate the binding strength using Eyring equation.

As cited in the new paragraph we added in the revision, we recently published a paper focused on the analysis of activator dissociation (Mizrak & Morgan, Nat. Comm. 2019; <https://www.nature.com/articles/s41467-019-13864-1>). This work made it clear that activator does not dissociate at a significant rate from the APC/C in purified complexes. This result is not surprising, as multiple cryoEM structures have shown that the interaction depends on extensive interactions between the activator N-terminal domain and a deep binding pocket on the APC/C.

(2) Do the authors only observe single-step photobleaching? Counting photobleaching steps may inform or confirm stoichiometry after controlling for the labelling efficiency.

As shown in the figures provided in our response to the first review, photobleaching occurs in a single step, and other methods cited in our response confirm that the great majority of the binding events we observe are single molecule binding events. Our dwell time analyses are most consistent with a single exponential decay, as expected for a single binding site. Finally, numerous cryoEM structures have demonstrated that the APC/C-activator complex contains a single binding site for each of the major degrons.